# Sexual selection in females and the evolution of polyandry

**Salomé Fromonteil**[1☯], **Lucas Marie-Orleach**[2,3☯], **Lennart Winkler**[4], **Tim Janicke**[1,4]*

**1** CEFE, Univ Montpellier, CNRS, EPHE, IRD, Montpellier, France, **2** Natural History Museum, University of Oslo, Oslo, Norway, **3** CNRS, Université de Rennes 1, ECOBIO (Écosystèmes, biodiversité, évolution)—UMR 6553, Rennes, France, **4** Applied Zoology, TU Dresden, Dresden, Germany

☯ These authors contributed equally to this work.
* tim.janicke@cefe.cnrs.fr

## Abstract

Over the last decades, the field of sexual selection underwent a paradigm shift from sexual-stereotype thinking of "eager" males and "coy" females towards a more nuanced perspective acknowledging that not only males but also females can benefit from multiple mating and compete for mating partners. Yet, sexual selection in females is still considered a peculiarity, and the evolution of polyandry is often viewed to result from a higher mating interest of males. Here, we present meta-analytic evidence from 77 species across a broad range of animal taxa to demonstrate that female reproductive success is overall positively correlated with mating success, suggesting that females typically benefit from multiple mating. Importantly, we found that these fitness gains likely promote the evolution of polyandry. Our findings offer support for the idea that sexual selection is widespread in females and to play a key role for the evolution of animal mating systems. Thereby, our results extend our understanding of the evolutionary consequences of sexual reproduction and contribute to a more balanced view of how sexual selection operates in males and females.

## Introduction

Sexual selection theory has become one of the most persuasive but also most controversial fields in evolutionary biology. Despite a general concord that sexual selection constitutes a potent evolutionary force shaping a great diversity of phenotypes in animals and plants [1], there is a continuing debate about the extent to which it operates differently in males and females [2–4]. When Darwin set the foundations of the field, he clearly considered males to be the primary target of sexual selection, which we here consider as selection arising from competition for mating partners and/or their gametes [5]. Specifically, Darwin argued that "with almost all animals, in which the sexes are separate, there is a constantly recurrent struggle between the males for the possession of the females" and that "the female [. . .], with the rarest exception, is less eager than the male [. . .,] she is coy and may often be seen endeavouring for a long time to escape from the male" [5]. Decades later, in a landmark contribution, Bateman speculated about the evolutionary causes of sex roles and cemented Darwin's sexual stereotypes through a series of arguments that later became known as Bateman's principles [6]. Most

**Funding:** This work was funded by the German Research Foundation (DFG grant number: JA 2653/2-1 to TJ) and the CNRS. The funders had no role in the study design, data collection and analysis, decision to publish, or preparation of the manuscript.

**Competing interests:** The authors have declared that no competing interests exist.

**Abbreviations:** GLMM, general linear mixed-effects model; HPD, Highest Posterior Density; REML, restricted maximum likelihood; UPGMA, unweighted pair group method with arithmetic mean.

importantly, he argued that the primordial sex difference in gamete size (i.e., small sperm versus big and nutrient-rich eggs) imposes sex-specific selection on mate acquisition, which eventually causes an "undiscriminating eagerness" in males and a "discriminating passivity" in females [6]. Thus, both founders of the field had the vision that sexual selection operates typically stronger on males compared to females—an assertion that has frequently been argued to have triggered an overly male-centred focus in research agendas of subsequent generations of evolutionary biologists [7].

Until today, sexual selection research on males predominates the field. In fact, studies testing for male–male competition and female choice outnumber those with a focus on female–female competition and male choice by magnitudes (Fig A in S1 Text). However, meta-analytic evidence suggests that Darwinian sex roles indeed prevail the animal tree of life [3]. This prompts the question to what extent the imbalance in research efforts reflects the aftermath of an alleged misconception by the pioneers [7–9] or whether the vast underrepresentation of studies on sexual selection in females has biological grounds because it corresponds to its rarity in nature [10,11].

Remarkably, neither Darwin nor Bateman ruled out that sexual selection operates in females. For example, Darwin argued that "In various classes of animals a few exceptional cases occur, in which the female instead of the male has acquired well-pronounced secondary sexual characters, such as brighter colours, greater size, strength, or pugnacity" [5]. Further, he acknowledged "With birds [. . .] there has sometimes been a complete transposition of the ordinary characters proper to each sex; the females having become the more eager in courtship, the males remaining comparatively passive, but apparently selecting, as we may infer from the results, the more attractive females." (p. 276). Thus, Darwin himself was the first to state that sexual selection can occur in females—an important but often overlooked implication of Darwin's pioneering work (but see [11]). Yet, only in the late 1990s, empiricists slowly began to accumulate evidence that females also compete for mating partners [11,12]. In fact, there is now multifaceted support for female–female competition and male choice at both pre- and postcopulatory episodes of sexual selection suggesting that sexual selection can act on females in a similar way as it does on males [13–17]. The most prominent and clearest support for sexual selection in females can be found in so-called sex-role reversed species in which females benefit relatively more from mating, and therefore often compete actively for males. For example, in some species of pipefishes and seahorses, fertilisation takes place inside the brood pouch of the male, which provides all parental care [18,19]. As a consequence, males become a limiting resource for which females compete, eventually leading to selection for ornaments favoured by male pre- and even postcopulatory mate choice [20]. Other examples of sex-role reversal are tropical shorebirds of the family Jacanidae in which females aggressively defend territories to monopolise multiple males [21]. Importantly however, sex-role reversal is not a prerequisite for sexual selection to operate in females, as it may represent just an extreme on a spectrum of sex roles. Even in species with Darwinian sex roles in which sexual selection promotes the evolution of male ornaments and extravagant courtship behaviours, females may still compete for access to high-quality males, as demonstrated in male lekking fruit flies [22] and peafowls [23]. Consequently, sexual selection in females might actually be an omnipresent phenomenon in animals but operating less intensely and more subtly compared to males [12,24].

In light of this development, there has clearly been a paradigm shift away from the sexual stereotypes dominating the early era of sexual selection research towards a more nuanced viewpoint acknowledging that females can be subject to sexual selection too. This progress was substantially fostered by the rise of molecular paternity analyses in the early 1990s revealing that females of many putatively monogamous species are actually polyandrous [25]—also

called the "polyandry revolution" [26], which spurred the quest for understanding the adaptive significance of multiple mating from a female perspective. Yet, the key question remains: Is polyandry primarily the consequence of a disproportionally higher eagerness to mate in males or are high levels of multiple mating also driven by a female interest [27–30]. The convenience-polyandry hypothesis posits that females engage in multiple mating not to obtain benefits but to limit costs imposed by male harassment [27]. Specifically, convenience polyandry is expected to occur if the cost of resistance to mate exceeds the net cost of mating. Interestingly, even if mating is associated with costs for fecundity and survival, polyandry has been demonstrated to be an evolutionary stable strategy if high mating rates reduce the risk of remaining unmated [30]. Moreover, irrespective of any mating costs, polyandry has been argued to evolve as a genetic corollary to sexual selection on males [31]. This hypothesis assumes a strong genetic correlation between male and female mating rates so that selection for a high mating propensity in males displaces females from their lower optimal mating rate.

In stark contrast to the "convenience-polyandry" and "genetic-corollary" hypotheses, polyandry has often been considered to evolve as a function of a female mating interest. Whenever the benefits of multiple mating outweigh the costs, selection on females is expected to favour a polyandrous mating system [32]. These benefits include so-called "direct" benefits (i.e., resources provided by males such as nuptial gifts, territory, or parental care), "indirect" or "genetic" benefits (i.e., if certain alleles or allele combinations increase offspring fitness), and benefits obtained from diversifying the genetic variation within a brood (i.e., genetic bet-hedging) [12]. There has been a tremendous effort in deciphering these potential benefits, and comparative studies on insects and birds suggest that females obtain primarily direct benefits [29], whereas meta-analytic evidence for indirect ("genetic") benefits is mixed [33–36]. Most importantly, explicit tests on whether the net benefit of multiple mating in females promotes the evolution of polyandry are virtually lacking. Ridley (1988) reviewed the literature on the benefits of mating in insects and found that the vast majority of studies reporting a fecundity increase with mating rate concerned polyandrous species [37]. Interestingly, Taylor and colleagues (2014) found that polyandry is common across a broad range of animal taxa but shows extensive intra- and interspecific variation [25]. They further found evidence for a weak correlation between the frequency of polyandry and multilocus heterozygosity (albeit not correcting for phylogenetic non-independence) suggesting that genetic benefits may contribute to the evolution of polyandry [25]. However, compelling comparative evidence for the evolution of polyandry in response to net benefits of multiple mating is missing.

Here, we aim at filling 2 major gaps in our understanding of sexual selection in females using a meta-analytic approach. First, we provide a quantitative assessment of the potential for sexual selection to operate in females across a broad range of animal taxa. Second, we test whether the net benefit of multiple mating, measured in terms of the Bateman gradient, predicts the evolution of polyandry across the animal tree of life as expected by the "benefits-driven" hypothesis. For these purposes, we compiled 120 published estimates of the so-called Bateman gradient, which measures the fitness benefit of mating. This metric captures the selective advantage arising from intra-sexual competition for mates, which is the core of Darwinian sexual selection [5,6]. Nevertheless, the Bateman gradient has a number of limitations that need to be taken into account for making reasonable interpretations (see Box 1). Importantly, Bateman gradients measure selection on mating success that is an important but not the only prerequisite for sexual selection to occur. Notably, access to mating partners also needs to be limited for which the Bateman gradient is largely silent (Box 1). Thus, we stress that the Bateman gradient is a proxy that quantifies the upper potential but not the actual strength of sexual selection.

## Box 1. What Bateman gradients tell about sexual selection and what they do not

In his landmark paper, Bateman aimed to unravel the ultimate reason for the sex difference in the strength of sexual selection as postulated by Darwin [6]. Inspired by an experiment with fruit flies, he argued that a stronger correlation between the number of mates and reproductive success observed in males is the *cause* of "intra-masculine" selection. Five decades later, Arnold and Duvall [32] formalised this idea by applying selection theory from quantitative genetics to provide a measure for the strength of sexual selection. They defined the *sexual selection gradient* ($\beta_{ss}$) as the slope of an ordinary least square regression of reproductive success (RS) on mating success (MS), which is

$$\beta_{ss} = \frac{cov(MS, RS)}{var(MS)}$$

In honour of Bateman's foundational work, $\beta_{ss}$ is often called the *Bateman gradient* [38] and advanced as a key metric to quantify the strength of Darwinian sexual selection. In essence, the Bateman gradient provides nothing else than an estimate of the fitness net return (i.e., benefits minus costs) that can be obtained from increasing mating success and therefore measures the strength of selection on mate acquisition. This implies that the Bateman gradient captures the selective advantage arising from intra-sexual competition for mates, which is the kernel of Darwinian sexual selection. In line with Bateman's original assertion, a positive Bateman gradient is predicted to promote competition for mates, determine the mating system, and favour the evolution of traits that confer a higher mating success such as ornaments and armaments [32,39]. The Bateman gradient does not provide a direct measure of selection on a sexually selected trait [40], but one important advantage is its eligibility to contrast the strength of sexual selection across contexts such as comparison among sexes, environments, and species [3,10,41]. However, for this purpose, the Bateman gradient needs to be computed on relativised data so that each individual estimate of reproductive success and mating success is divided by the mean value of the given sample [42]. Another critical asset of the Bateman gradient is that it addresses a testable hypothesis (i.e., reproductive success is related to mating success), which makes it the only sexual selection metric that can serve as an effect size in meta-analyses.

Despite its capacity to provide a universal proxy for the strength of sexual selection, the Bateman gradient has a number of limitations, which need to be taken into account for reasonable interpretation. One important conceptual drawback of the Bateman gradient is that, at best, it only informs about the net return for obtaining a (additional) mate, which is an important but not the sole prerequisite for intra-sexual competition to arise. Specifically, for competition to occur, mating partners also have to be a limited resource [43]. Only if mate acquisition is difficult (i.e., costly), individuals compete for mates and sexual selection can operate. Mate limitation can have many reasons including low densities and skewed operational sex ratios but always implies that some individuals cannot achieve an optimal mating success, which translates into variance in mating success. Remarkably, already Bateman denoted the variance in mating success as a *sign* for "intra-masculine" selection, which later became a proxy for the intensity of precopulatory competition (i.e., termed the "opportunity for sexual selection;" $I_S$) by providing an upper limit for the strength of directional sexual selection (for a critical review see [40]).

Importantly, the absence of variance in mating success implies an absence of competition for access to mating partners [42]. This corresponds to the very basic theorem of selection theory: selection requires not only a fitness effect of a trait but also variance in that trait. For an evolutionary response, at least some fraction of that variance needs to have a heritable basis. With respect to Darwinian sexual selection, the trait of interest is mating success and the Bateman gradient informs about its fitness effect but not about its variance. Jones [42] combined both components of sexual selection in a single metric as

$$s'_{max} = \beta_{SS}\sqrt{I_S},$$

where $s'_{max}$ is termed the maximum standardised sexual selection differential or simply the *Jones' index*. This metric estimates the maximum strength of precopulatory sexual selection on a trait and has been demonstrated to outperform $\beta_{SS}$ and $I_S$ in a simulation study [44].

Another shortcoming of the Bateman gradient is its focus on premating sexual selection, which was the focus of sexual selection envisioned by Darwin. Yet, postmating sexual selection in terms of postcopulatory competition and choice have been identified as major components of sexual selection [45]. This can be a significant constraint for measuring sexual selection in males for which sperm competition has been found to be intense in a broad array of species [46]. By contrast, egg competition seems to be rare and restricted to external fertilizers [13], which makes the female Bateman gradient a less incomplete proxy for the total strength of sexual selection compared to males.

Apart from these conceptual limitations of Bateman gradients, there are a number of methodological aspects that need to be taken into account when interpreting Bateman gradients. First and foremost, like most selection differentials, the vast majority of Bateman gradients rely on correlational data so that they do not allow inference of causality. Thus, a positive relationship between mating success and reproductive success can be cofounded with other unmeasured factors such as body size. Moreover, the actual causal relationship can be inversed such that reproductive success affects mating success due to a preference for mating with more fecund partners [47]. This confounding can be problematic for female Bateman gradients given the ubiquitous evidence that males can be choosy with respect to the partner's fecundity [17]. Second, the explanatory power of the Bateman gradient depends on how reproductive success and mating success are estimated. Especially the measurement of actual mating success based on behavioural observations can be very laborious and sometimes even impossible. Therefore, mating success is often inferred from genetic parentage analysis and defined as the total number of mates with whom an individual produced offspring (i.e., genetic mating success). A meta-analysis proved that this approach inflates the Bateman gradient of males and females when compared to estimates derived from behavioural measures of mating success (i.e., copulatory mating success) [48]. Quantifying mating success in terms of the number of genetic parents may not only obscure a potentially important component of postcopulatory sexual selection (because unsuccessful copulations and multiple copulations with the same partner remain undetected) but also leads to an autocorrelation of mating success and reproductive success, especially in species with low fecundity [49,50].

Finally, another notable limitation of the Bateman gradient is its assumption of a linear relationship between mating success and reproductive success, which can be an oversimplification, especially in females. Specifically, in separate-sexed species, reproductive success of individuals with no mating success is necessarily zero but increases as soon as one successful mating is obtained. However, after 1 mating, reproductive success may further increase, remain constant or even decrease with further matings (e.g., if mating entails cost associated with harm or transmission of diseases). Consequently, the relationship between reproductive success and mating success may become nonlinear if fitness is optimised at intermediate mating rates, which has been demonstrated for females in many species [51]. For this reason, it is often informative to compute Bateman gradients with and without individuals having zero mating success, with the latter quantifying the fitness return of an additional mating [48].

More detailed reflections on the strengths and limitations of the Bateman gradient can be found elsewhere including guidelines on how to avoid pitfalls in estimating Bateman gradients and how to control for potential confounding factors [48,52,53].

## Results

We found evidence for a high potential of sexual selection to operate in females across the animal tree of life in terms of a positive global effect size of the Bateman gradient (Fig 1A and Table 1 and Table A in S1 Text). Our phylogenetically independent meta-analysis revealed a significant phylogenetic signal (phylogenetic heritability $H^2$ = 0.42; Table 1), which is also reflected in differences among major taxonomic groups with effect sizes being highest in fish (Table B in S1 Text). Moreover, Bateman gradients showed substantial variability across studies (Fig 2 and Table 1). This variation was partly explained by differences in methodological approaches used to quantify the strength of sexual selection. Specifically, estimates of sexual selection critically depended on how mating success was measured (Table 2 and Table C in S1 Text and Fig B in S1 Text): higher effect sizes were observed in studies using genetic parentage analysis to assess mating success (i.e., genetic mating success) compared to estimates based on behavioural observations (i.e., copulatory mating success). In addition, inclusion of individuals that did not mate led to larger effect sizes compared to estimates excluding individuals with zero mating success (Table 2 and Table C in S1 Text and Fig B in S1 Text). Nonetheless, we still observed a signal for positive selection on mating success when running more conservative analyses restricted to studies relying on copulatory mating success or studies excluding individuals that did not mate (Table 1 and Table A in S1 Text). Furthermore, we did not detect a significant difference in female Bateman gradients between laboratory and field studies (Table 2 and Table C in S1 Text).

Even though none of the sampled species showed strict monogamy, they differed considerably in the level of polyandry quantified as the proportion of females in the population with more than one mating partner (mean ± SE = 0.66 ± 0.03; range = 0.01–1.00). Remarkably, this interspecific variation in the mating system was related to the Bateman gradient. Species that were more polyandrous showed steeper Bateman gradients, regardless of whether polyandry was considered as discrete categories of low- versus high-polyandry (Table 2 and Table C in S1 Text and Fig 1B) or as a continuous variable (Table 2 and Table C in S1 Text).

We detected no signature for publication bias based on multilevel meta-regression testing for a relationship between effect size and its standard error (GLMM: estimate ± SE, −0.135 ± 0.478, $P_{MCMC}$ = 0.778; Fig C in S1 Text). Finally, we did not detect an effect of

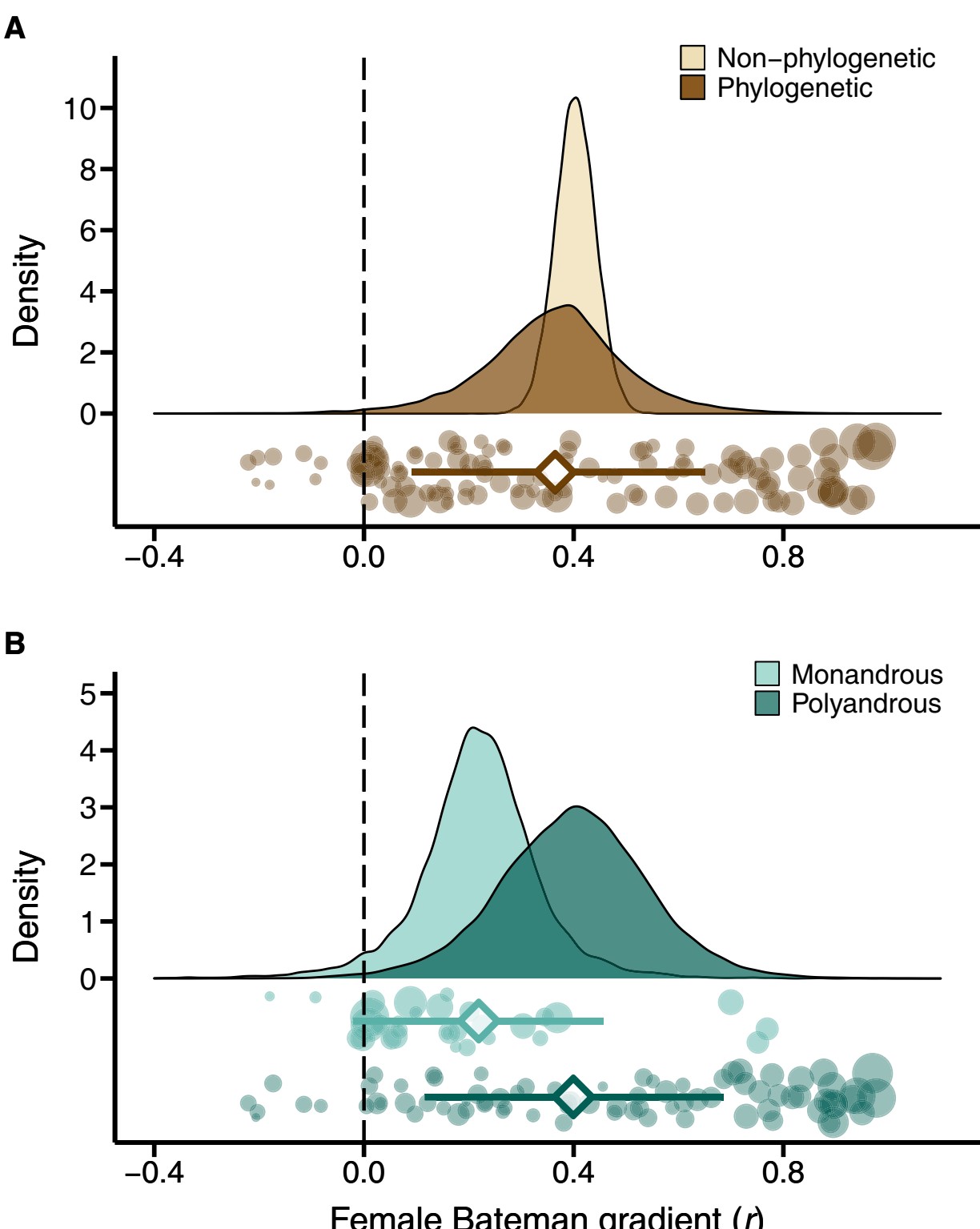

**Fig 1. Meta-analytic evidence for sexual selection in females and its relation to the mating system.** (A) Global effect size of the Bateman gradient obtained from GLMMs with or without accounting for phylogenetic non-independence (phylogenetic or non-phylogenetic, respectively). (B) Contrast in sexual selection in females between low-polyandry and high-polyandry species. Raincloud charts show posterior distributions, global effect size with 95% HPD intervals (diamonds and error bars) and raw effect sizes (filled circles) of female Bateman gradient. The code and data needed to generate this figure can be found at https://salomefromonteil.github.io/META_SexSelFem/ and https://doi.org/10.5281/zenodo.7303598. GLMM, general linear mixed-effects model; HPD, Highest Posterior Density.

**Table 1. Global tests of sexual selection in females.**

| Model | $k$ | $N_{Species}$ | Effect size | | | | Heterogeneity | | | | | | | | |
|---|---|---|---|---|---|---|---|---|---|---|---|---|---|---|---|
| | | | $r$ | | | $P_{MCMC}$ | $I^2_{Phylogeny}$ | | | $I^2_{Study}$ | | | $I^2_{Observation}$ | | |
| Global model (non-phylogenetic) | 120 | 77 | 0.41 | (0.34, | 0.48) | < 0.001 | - | | | - | | | - | | |
| Global model (phylogenetic) | 120 | 77 | 0.38 | (0.14, | 0.61) | 0.006 | 0.42 | (0.01, | 0.80) | 0.40 | (0.05, | 0.79) | 0.09 | (0.00, | 0.24) |
| Copulatory mating success | 43 | 24 | 0.23 | (0.05, | 0.40) | 0.016 | 0.37 | (0.01, | 0.79) | 0.15 | (0.00, | 0.45) | 0.25 | (0.00, | 0.64) |
| Genetic mating success | 79 | 56 | 0.50 | (0.29, | 0.70) | < 0.001 | 0.26 | (0.00, | 0.67) | 0.56 | (0.15, | 0.90) | 0.09 | (0.00, | 0.25) |
| Including zero mating success | 70 | 42 | 0.43 | (0.15, | 0.69) | 0.006 | 0.48 | (0.00, | 0.85) | 0.29 | (0.00, | 0.70) | 0.12 | (0.00, | 0.35) |
| Excluding zero mating success | 79 | 58 | 0.33 | (0.14, | 0.51) | 0.005 | 0.29 | (0.00, | 0.73) | 0.58 | (0.16, | 0.92) | 0.06 | (0.00, | 0.19) |
| Laboratory studies | 52 | 31 | 0.40 | (0.14, | 0.66) | 0.007 | 0.57 | (0.11, | 0.92) | 0.15 | (0.00, | 0.51) | 0.14 | (0.00, | 0.41) |
| Field studies | 68 | 47 | 0.37 | (0.10, | 0.62) | 0.022 | 0.27 | (0.00, | 0.80) | 0.60 | (0.10, | 0.94) | 0.06 | (0.00, | 0.19) |
| Low-polyandry species | 32 | 16 | 0.22 | (-0.02, | 0.45) | 0.065 | 0.24 | (0.00, | 0.68) | 0.53 | (0.01, | 0.91) | 0.12 | (0.00, | 0.47) |
| High-polyandry species | 88 | 61 | 0.41 | (0.15, | 0.66) | 0.004 | 0.64 | (0.27, | 0.91) | 0.13 | (0.00, | 0.41) | 0.11 | (0.00, | 0.31) |

Results of intercept-only phylogenetically controlled GLMMs are shown for the entire dataset (global model) and subsets with respect to mating success method (copulatory versus genetic), mating success range (including versus excluding zero mating success category), study type (laboratory versus field studies), and mating system (low-polyandry versus high-polyandry species). Table shows number of effect sizes ($k$), number of species ($N$), effect size ($r$), and heterogeneity $I^2$ arising from phylogenetic affinities, between-study variation, and between-observation variation. Model estimates are shown as posterior modes with 95% HPD intervals in parentheses.

GLMM, general linear mixed-effects model; HPD, Highest Posterior Density.

publication year suggesting the absence of the so-called bandwagon effect [54] (Table 2 and Table C in S1 Text).

## Discussion

The field of sexual selection underwent a paradigm shift from stereotypic sex-role thinking toward a less biased perspective on how competition for mating partners and their gametes imposes selection on both sexes. In agreement with the pioneering work by Darwin and Bateman, sexual selection has been found to act more strongly on males than on females. However, does this sex difference preclude sexual selection to be widespread in females? Moreover, if sexual selection in females is frequent, does it contribute to the diversity of mating systems? Here, we provide meta-analytic evidence suggesting that sexual selection is potentially common in female animals and that the benefits of multiple mating in females—approximated by the Bateman gradients—explain the variation in polyandry across the animal tree of life.

Our phylogenetically informed synthesis suggests that females—just as males—typically benefit from having more than one mating partner. Therefore, our study offers quantitative evidence that positive selection for mate acquisition may potentially be common in females, which is expected to favour the evolution of sexual traits in females across a broad taxonomic range, and may therefore challenge arguments that ornamentation in females evolves mainly as a by-product of sexual selection on males [55]. However, a positive Bateman gradient in females, alone, may not suffice to promote the evolution of female sexually selected traits. Especially in species in which female Bateman gradients are positive but less steep compared to males, female ornaments and armaments may not evolve because males may not be a limiting resource given their even higher selective advantage of being polygamous. Hence, the apparent underrepresentation of sexually selected traits in females observed across animals does not necessarily contradict the overall benefit of multiple mating in females. Interestingly, however, in all compiled primary studies female mating success varied among individuals

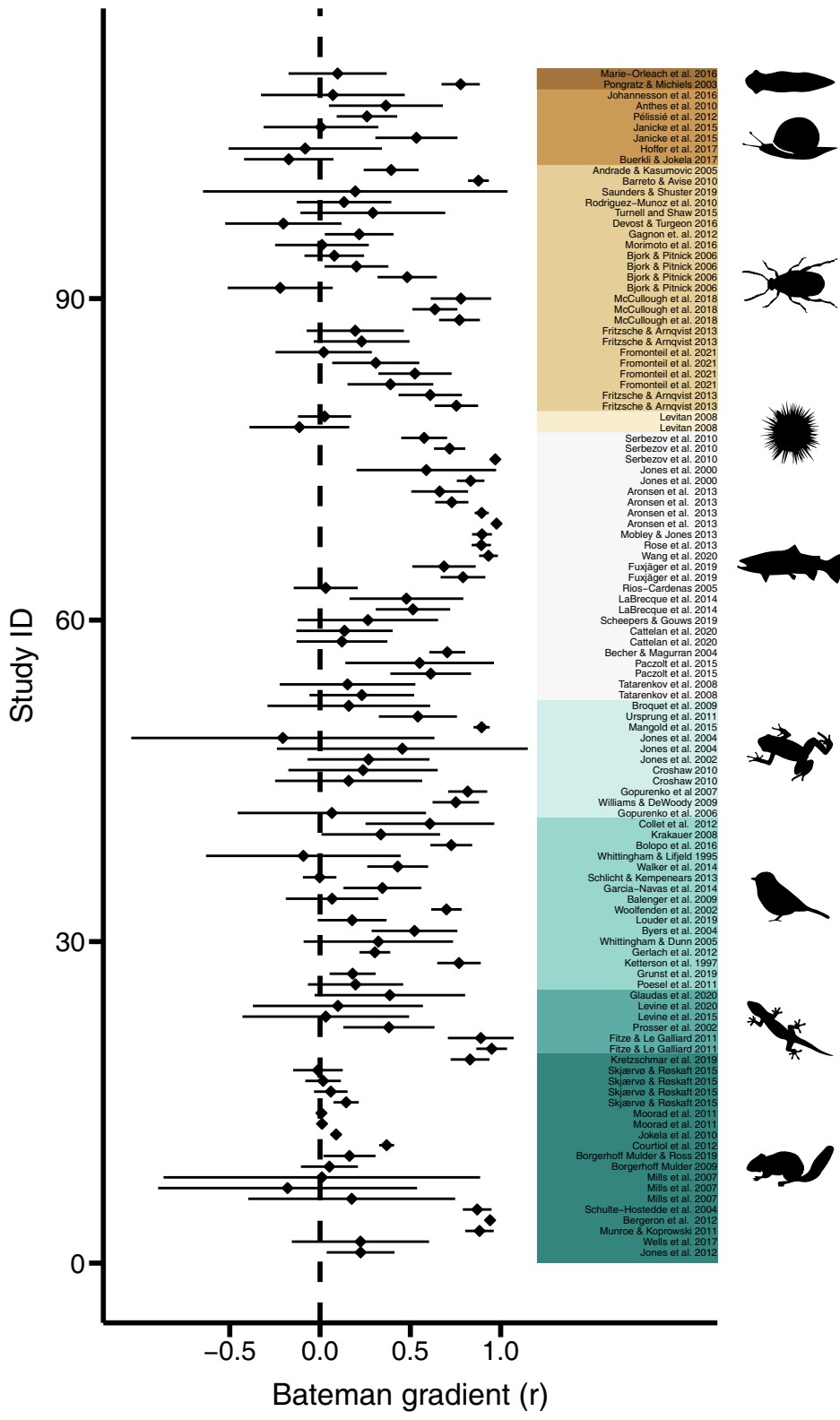

**Fig 2. Forest plot.** Graphs shows all sampled effect sizes (Pearson correlation coefficient of Bateman gradients) with 95% confidence limits in phylogenetic order. The code and data needed to generate this figure can be found at https://salomefromonteil.github.io/META_SexSelFem/ and https://doi.org/10.5281/zenodo.7303598.

**Table 2. Predictors of interspecific variation in female Bateman gradients.**

| Moderator | Estimate | | | $P_{MCMC}$ | $R^2$ | | |
|---|---|---|---|---|---|---|---|
| Mating success method | 0.33 | (0.18, | 0.49) | <0.001 | 0.22 | (0.14, | 0.30) |
| Mating success range | 0.12 | (0.05, | 0.19) | 0.002 | 0.03 | (0.01, | 0.04) |
| Study type | 0.06 | (−0.11, | 0.25) | 0.481 | 0.01 | (0.00, | 0.01) |
| Year of publication | 0.00 | (−0.02, | 0.01) | 0.427 | 0.01 | (0.00, | 0.01) |
| Mating system | 0.31 | (0.16, | 0.46) | <0.001 | 0.15 | (0.08, | 0.22) |
| Polyandry | 0.66 | (0.37, | 0.94) | <0.001 | 0.18 | (0.10, | 0.27) |

Methodological moderators include mating success method (copulatory versus genetic mating success), mating success range (including versus excluding zero mating success category), study type (field versus lab), and year of publication (continuous variable). Effect of mating system contrasts low-polyandry and high-polyandry species. Effect of polyandry estimates the relationship between the female Bateman gradient and the proportion of polyandrous females in the population. Model estimates (i.e., estimated difference between groups) are shown as posterior modes with 95% HPD intervals obtained from phylogenetically controlled GLMMs. The variance explained by the moderator variable is given as the marginal $R^2$ with 95% HPD intervals in parentheses.

GLMM, general linear mixed-effects model; HPD, Highest Posterior Density.

because otherwise, the authors would not have been able to quantify the relationship between reproductive success and mating success. This suggests that in all tested systems for which the Bateman gradient was found to differ from zero, at least some females did not achieve their optimal mating success (assuming that females of a given species have a shared optimum). Admittedly, variance in mating success can have many causes including variance in another correlated trait, stochasticity, and/or could just be an artefact due to the experimental conditions used to assess Bateman gradients (e.g., the amount of time in which females were allowed to interact with males in laboratory studies). Yet, a large fraction of the compiled published work concerns field studies (i.e., 55 out of 84; 65.5%) of which many focus on open populations and span over an entire reproductive season or even lifetime. Consequently, the variance in female mating success observed in the majority of primary studies is not driven by artificial experimental conditions but may instead reflect to some degree that males are a limited resource for females. Nonetheless, the extent to which variance in female mating success is indicative of females being male-limited, or whether it results from stochasticity or from variance in another correlated trait, remains an interesting question for future empirical work, especially in species with nonzero Bateman gradients.

The other major finding of our study is that species with a positive female Bateman gradient tend to be more polyandrous, which has long been argued [32] but, to our knowledge, has never been tested across species. Even if our comparative approach does not allow inference of causality, this result suggests that positive selection on mating success in females translates into higher mating rates as predicted by sexual selection theory [12,32]. Hence, our results support the hypothesis that the evolution of polyandry is facilitated when females benefit from multiple mating, and thus, refute alternative hypotheses in which the evolution of polyandry is assumed to be male-driven and evolves primarily to mitigate costs associated with mating ("convenience polyandry" hypothesis; [27]) or because of a genetic corollary to sexual selection on males [31].

Our study relies on the premise that the Bateman gradient provides a meaningful quantitative proxy for the strength of sexual selection. While there is compelling theoretical and empirical support for this assertion, especially in the context of interspecific comparisons [10,42,44,48], the Bateman gradient has a number of limitations (Box 1). Presumably, the most critical shortcoming is that Bateman gradients, like most selection gradients, are typically inferred from descriptive approaches in which the predictor variable (i.e., mating success) is

not manipulated experimentally. Thus, Bateman gradients do not imply causality because a positive relationship between mating success and reproductive success in females can either indicate an actual fitness benefit of mating or that fecundity affects mating success (e.g., due to a male preference [47]). As another limitation, the Bateman gradient only captures the upper potential of actual phenotypic selection [48], which implies that our study cannot provide a trait-based perspective on female sexual selection. More specifically, Bateman gradients do not quantify the costs associated with the development of a phenotypic trait value, which allows to achieve an additional mating [43]. Finally, Bateman gradients may also underestimate the strength of sexual selection because they focus only on the number of partners or copulations as the target of selection. For example, when sexual selection involves competition for mate quality rather than quantity, which might be particularly relevant for females [56], Bateman gradients are incomplete estimates of the strength of sexual selection.

Despite these limitations, our results are robust with respect to different methodological approaches used to quantify the Bateman gradient. In our study, even after the exclusion of study designs that are prone to overestimate the relationship between mating and reproductive success, we detected an overall positive Bateman gradient. Specifically, studies inferring mating success from parentage (i.e., genetic mating success) have been shown to overestimate the Bateman gradient [48,57]. However, when we restrict our analysis to studies in which mating success was measured on behavioural observations (i.e., copulatory mating success), we still found a positive global effect size. In addition, the likelihood to detect multiple sires increases with female fecundity, which may lead to an autocorrelation between female mating success and female reproductive success in studies relying on genetic mating success [48]. This spurious relationship did not seem to have formed the basis of our observed effects, since we did not detect a correlation between estimates of Bateman gradients and female fecundity in studies using genetic mating success. Moreover, our findings suggest that the positive relationship between mating success and reproductive success in females is not only driven by the benefit of having at least a single mating but also by the benefit of having an additional mating. Hence, despite various lines of evidence that mating can incur costs for females [58,59], our data suggest that reproductive success may often be maximised at high mating rates.

Collectively, our study contributes to a more nuanced view on sexual selection and sex differences in general. Although Darwinian sex roles seem to predominate the animal tree of life in the sense that sexual selection is typically stronger on males compared to females [3], our meta-analysis corroborates the often alleged but hitherto untested assumption that sexual selection can be an important evolutionary force in females shaping animal mating systems. Ultimately, our findings prompt the question of whether females of species with positive Bateman gradients only accept more mating attempts by males and therefore become more polyandrous or whether they actively strive and compete for more mating opportunities. Given the mentioned limitations of Bateman gradients, our study can only reveal a high potential for sexual selection to be widespread in females but may mark a starting point for further empirical research exploring actual female–female competition for mating partners and/or gametes in species characterised by positive female Bateman gradients and Darwinian sex roles. Moreover, positive Bateman gradients in females may weaken selection on sexual traits in males because females may become less choosy, which may also relax sexual conflict over mating. Yet, our current knowledge on how sexual selection in one sex affects sexual selection in the other is very limited. Detailed knowledge of such interactions is clearly pivotal to better understand intra- and interspecific variation in the strength of sexual selection and represents a promising avenue for future theoretical and empirical work on the evolution of mating systems.

## Materials and methods

### Systematic literature search

We extracted female Bateman gradients from a previous meta-analysis [3] and expanded this database by adding studies that have since been published. Specifically, we ran a systematic literature search using the ISI Web of Knowledge (ISI Web of Science Core Collection database; Clarivate Analytics) with the "topic" search terms defined as ("Bateman*" OR "opportunit* for selection" OR "opportunit* for sexual selection" OR "selection gradient*" OR ("mating success" AND "female*")) on the 31st of March 2022. In this search, the timespan was defined as "2015 – today" because the literature search of the previous study had been carried out on the 25th of April 2015. In addition, we also screened all studies published after 2015 that cited Bateman's original paper. Our sole inclusion criterion was that the study must report data allowing to assess the relationship between mating success and reproductive success for females. The search yielded 1,974 records of which 30 studies were considered eligible, providing a total of 39 additional estimates of female Bateman gradients. In addition, we included 4 estimates from an unpublished experimental study on the bean weevil *Acanthoscelides obtectus* (S. Fromonteil and colleagues, unpublished data) and 4 estimates obtained from a study on the red flour beetle *Tribolium castaneum* (L. Winkler and colleagues, unpublished data). Combining these estimates with the ones obtained from the previous meta-analysis added up to a final dataset of 84 studies reporting 120 female Bateman gradients from 77 species (Fig 3 and S2 Text).

### Moderator variables

Apart from a global test of sexual selection in females (inferred from a positive Bateman gradient), we aimed at explaining among-study variation in effect sizes from both a methodological and an evolutionary perspective. First, we evaluated if the method to quantify mating success influenced Bateman gradients. Especially for females, the measurement of mating success in terms of the number of genetic partners (i.e., genetic mating success) has been demonstrated repeatedly to overestimate the Bateman gradient when compared to estimates obtained from behavioural observations (i.e., copulatory mating success) [48]. Quantifying mating success in terms of the number of genetic partners may not only obscure a potentially important component of postcopulatory sexual selection (because unsuccessful copulations and multiple copulations with the same partner remain undetected) but also leads to an autocorrelation of mating success and reproductive success, particularly in species with low fecundity [50]. For those reasons, we tested the effect of the mating success method by contrasting estimates of Bateman gradients based on genetic ($k = 77$) versus copulatory mating success ($k = 43$). Studies using copulatory mating success relied on behavioural observations of the actual number of copulatory partners ($k = 28$) or the total number of copulations ($k = 15$). Second, we explored the impact of having unmated individuals included in the measurement of the Bateman gradient. Estimates including this zero-mating success category provide a combined estimate for the benefit of mating once and the benefit of having an additional mating partner (or copulation), whereas Bateman gradients excluding zero-mating success data capture only the latter. In the context of sexual selection, we are primarily interested in the benefit of having an additional mating partner (or copulation) rather than the benefit of mating itself, since the latter is essential for reproduction in outcrossing species. Thus, we compared Bateman gradients that include unmated individuals ($k = 70$) with those excluding this zero-mating success category ($k = 79$). Third, to further account for methodological differences between studies, we tested for an effect of the study type on Bateman gradients by comparing field studies ($k = 68$) with laboratory studies ($k = 52$).

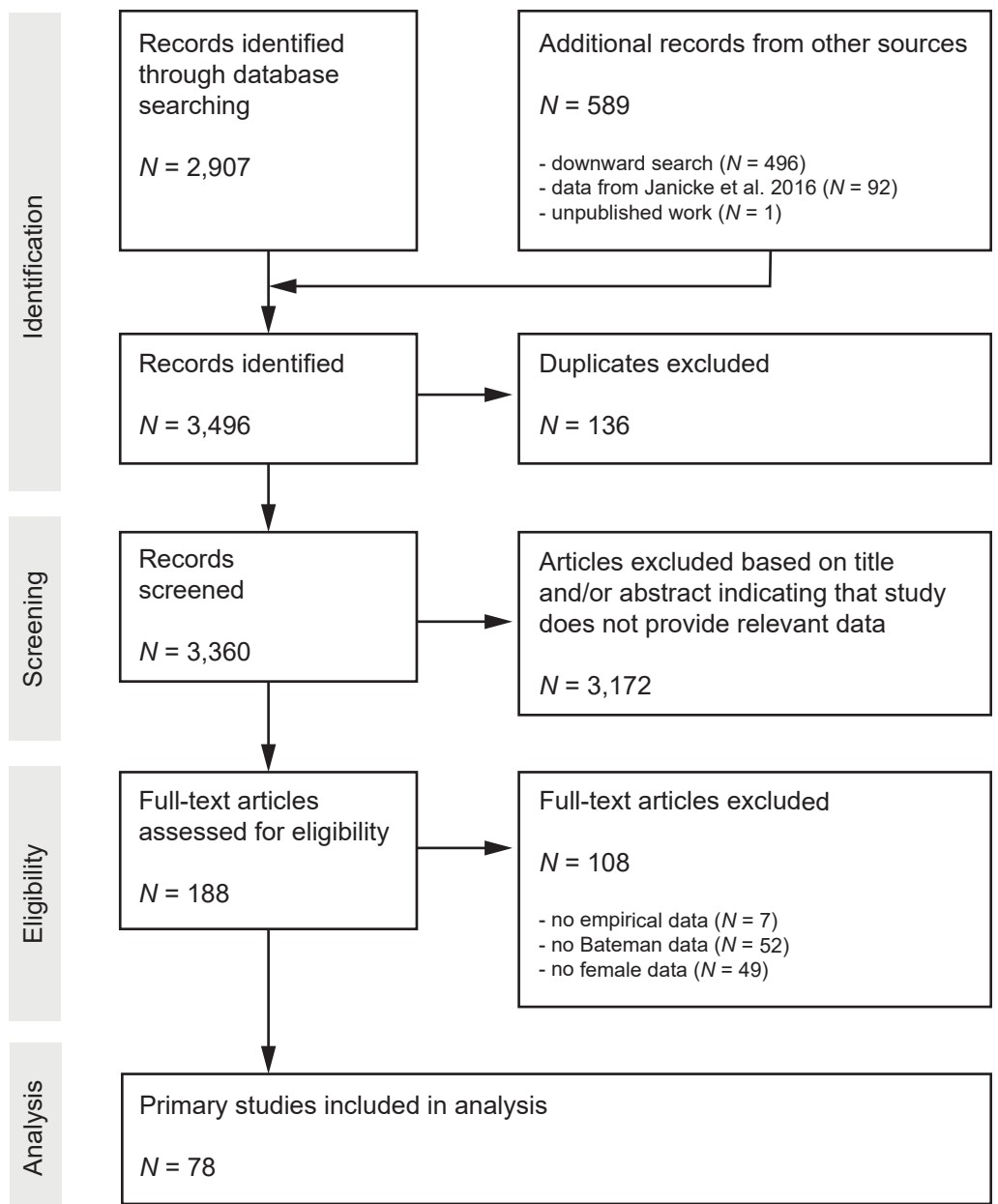

**Fig 3. Preferred Reporting Items for Systematic Reviews and Meta-Analyses (PRISMA) Diagram.** Flow chart maps the number of records identified during the different phases of the systematic literature search.

Fourth, we tested whether the Bateman gradient was related to mating system. We predicted that a fitness benefit of achieving high mating success selects for increased polyandry, meaning that species with a stronger female Bateman gradient are expected to be more polyandrous [32]. We classified the mating system of each sampled species based on estimates of polyandry, which we defined as the proportion of reproducing females that have more than 1 mating partner. For the majority of species ($N = 66$; 85.7%), we estimated the proportion of multiply mated females using data provided in the primary studies (Table D in S1 Text). For most of the remaining species, we extracted estimates of polyandry from secondary literature, except for 3 species for which we could only find verbal classifications of the mating system

(see Table D in S1 Text for references). We then used these estimates to define the mating system as either low-polyandry or high-polyandry, depending on whether its value was lower or higher than 0.5, respectively, because this value has been found to be the average level of polyandry in wild populations [25]. In total, our dataset encompassed 16 high-polyandry and 61 low-polyandry species, for which we obtained 32 and 88 effect sizes, respectively. Sensitivity analyses revealed that alternative thresholds of polyandry (i.e., 0.4 or 0.6) did not lead to qualitative changes of results. We note that our classification of the mating system remains an oversimplification of a clearly more gradual spectrum of natural mating systems. Unfortunately, an alternative model in which we used the actual estimate of polyandry as a continuous predictor variable showed significant heteroscedasticity (studentised Breusch–Pagan test: $\chi^2 = 7.775$, df = 1, $P = 0.005$). Therefore, we prefer to base our conclusions on the model including mating system as a binary factor, but for completeness, we also report the outcome of the alternative model.

## Phylogenetic affinities

We reconstructed the phylogeny of all sampled species from published data in order to account for phylogenetic non-independence (Fig D in S1 Text). Specifically, we extracted divergence times from the TimeTree database (http://www.timetree.org/; [60]) and transformed the distance matrix into the NEWICK format using the unweighted pair group method with arithmetic mean (UPGMA) algorithm implemented in MEGA (https://www.megasoftware.net/; [61]). In total, our analysis included 77 species with a broad distribution across the animal tree of life, with an overrepresentation of arthropods ($N_{Species} = 20$), birds ($N_{Species} = 13$), fishes ($N_{Species} = 15$), and mammals ($N_{Species} = 8$) (Fig D in S1 Text).

## Statistical analysis

The Bateman gradient is defined as the slope of a linear regression of reproductive success on mating success [6] and provides a powerful metric of the strength of sexual selection for interspecific comparisons when computed on relativised data (i.e., accounting for differences in mean mating and reproductive success) [42]. However, only 61.7% of the extracted Bateman gradients were computed on relativised data. Therefore, we converted all obtained slopes into Pearson correlation coefficients ($r$) and computed their sampling variances using formulas reported elsewhere [62]. We note that using $r$ as an effect size instead of a slope quantifies the strength of the relationship between mating success and reproductive success, which depends not only on the slope (i.e., the fitness return of the mating) but also on the goodness of fit (i.e., the standard error of the slope). However, analysis of the subset of data for which we could extract standardised Bateman gradients revealed that $r$ is a strong predictor of the actual Bateman gradient (Linear Regression: estimate ± SE = 1.19 ± 0.06; $F_{1,72} = 375.88$; $P < 0.001$, $R^2 = 0.84$; Fig E in S1 Text), suggesting that our effect size is a reliable estimate for the benefit of mating.

Even though the Bateman gradient is a well-established metric for interspecific comparisons of the strength of sexual selection, it has various limitations (see Box 1 for a critical account) and alternative metrics have been proposed to quantify sexual selection. Most and foremost, the maximum standardised sexual selection differential $s'_{max}$ (i.e., the product of the standardised Bateman gradient and the square root of the variance in relativised mating success $I_s$; [42]) has been found to outperform the Bateman gradient, especially when measuring sexual selection in females [44]. Given that only a fraction of primary studies reported Bateman gradients on relativized data (see above), we could not use $s'_{max}$ as target response variable. However, an analysis restricted to primary studies reporting both standardised Bateman

gradients and estimates of $I_s$ ($N = 73$) suggests that our effect size $r$ is a good predictor of $s'max$ (Linear Regression: estimate ± SE = 1.00 ± 0.09; $F_{1,71} = 116.3$; $P < 0.001$, $R^2 = 0.62$; Fig E in S1 Text). Finally, studies in which mating success is inferred from genetic parentage have been argued to result in spurious Bateman gradients mainly due to an autocorrelation between predictor and response variable, which is expected to be especially problematic for species with an overall low female fecundity [57]. If this imposes a major bias in our data, we would predict that Bateman gradients increase with decreasing female fecundity. However, we did not find evidence for a significant negative relationship between female fecundity and effect sizes of Bateman gradients estimated from genetic parentage (Linear Regression: estimate ± SE = −-0.05 ± 0.06; $F_{1,77} = 0.773$; $P = 0.382$, $R^2 = 0.01$), which suggests that positive Bateman gradients observed in those studies are not only driven by the mentioned autocorrelation.

We ran general linear mixed-effects models (GLMMs) to provide a global test for sexual selection in females and to explore determinants of the inter-study variation. First, we quantified global effect sizes by running GLMMs with $r$ defined as the response variable weighted by the inverse of its sampling variance and included study identifier and observation identifier as a random term. This was done both without (i.e., "non-phylogenetic" GLMMs) and with adding the phylogenetic correlation matrix as an additional random term ("phylogenetic" GLMMs). Secondly, we ran phylogenetic GLMMs in which we defined mating success method (copulatory versus genetic), mating success range (with versus without zero-mating success category), study type (field versus laboratory studies), or mating system as a fixed factor to explain inter-study variation in $r$. In order to complement our analysis of the mating system, we also ran a phylogenetic GLMM including estimates of the actual level of polyandry (i.e., the proportion of multiply mated females) as a continuous predictor variable. All GLMMs were run with the MCMCglmm function of the MCMCglmm R package version 2.29 [63], using uninformative priors ($V = 1$, $nu = 0.002$) and an effective sample size of 10,000 (number of iterations = 4,400,000, burn-in = 400,000, thinning interval = 400). All models were also run with alternative priors, which revealed qualitatively identical results. Moreover, we ran all models multiple times to verify convergence and checked for autocorrelation in the chains. For completeness, we also ran all GLMMs using the restricted maximum likelihood (REML) approach using the metafor R package version 2.4–0 [64]. These complementary analyses provided qualitatively similar results and are reported in the Supporting information (Tables A and C in S1 Text).

We estimated heterogeneity $I^2$ from the intercept-only model as the proportion of variance in effect size that can be attributed to the different levels of random effects [65]. In particular, we decomposed total heterogeneity into the proportional phylogenetic variance ($I^2_{Phylogeny}$), between-study variance ($I^2_{Study}$), and study-specific variance (observation-level random effect; $I^2_{Observation}$) [66]. Note that $I^2_{Phylogeny}$ is also termed phylogenetic heritability $H^2$ and is equivalent to Pagel's λ [67]. For models including predictor variables, we computed the proportion of variance explained by those fixed factors ("marginal $R^2$") [68].

We used multilevel meta-regression to explore the potential for publication bias [69]. We first transformed our effect size $r$ into Fisher's $z$ statistics and computed its variance using formulas reported elsewhere [70]. This was done because the sampling variance of $z$ only depends on the sample size but not on the effect size itself, which is not the case for Pearson's correlation coefficient. We then tested whether the effect size depends on its standard error, which may suggest that small studies only get published if effect sizes are large enough to provide statistically significant support for the tested hypothesis. Specifically, we ran a GLMM with $z$ defined as response variable, its standard error as fixed effect and study identifier, observation identifier and the phylogenetic correlation matrix as random terms. Moreover, we tested whether the year of publication influences effect sizes, which has been argued to be suggestive

of other forms of biases [54]. For example, the so-called bandwagon effect suggests that supportive results get easier published in a newly emerging field but over time scepticism about the theoretical foundations may arise and initially non-intuitive findings may find a more receptive audience. If true for the field of sexual selection, we may expect an increase of effect sizes for female Bateman gradients with the rising awareness in the community that sexual selection does not only operate in males.

Some readers might wonder whether so-called sex-role reversed species are overrepresented in our dataset because species in which females are known to compete for males have repeatedly been studied to provide a proof of concept of Bateman's principles. Moreover, the inclusion of human studies in our meta-analysis might be problematic for at least 2 reasons. First, mating success in human studies is often estimated in terms of number of pair bonds or marriages, which might be very different from the actual number of sexual partners. Second, the estimated level of polyandry in humans is the lowest among all species in our analysis (outlier analysis: $\chi^2 = 8.367$, $P = 0.004$), which may bias the tested relationship between the female Bateman gradient and the level of polyandry. For those reasons, we ran an additional series of analyses excluding sex-role reversed species or estimates obtained from human studies. These analyses suggest that all results obtained from the analysis of the complete dataset remain robust after excluding sex-role reversed species (Tables E and F in S1 Text) or human studies (Tables G and H in S1 Text).

All statistical analyses were carried out in R version 4.0.3 [71] and all data together with R scripts used to perform the presented analyses have been made available online at Zenodo (https://doi.org/10.5281/zenodo.7303598) and GitHub (https://salomefromonteil.github.io/META_SexSelFem/).

## Supporting information

**S1 Text. Supplementary information. Fig A.** Imbalance between studies of sexual selection in males and females. **Fig B.** Methodological predictors of female Bateman gradients. **Fig C.** Egger's regression. **Fig D.** Phylogenetic tree of all sampled species. **Fig E.** Significance of the used effect size. **Table A.** Global tests of sexual selection in females using restricted maximum likelihood REML approach. **Table B.** Comparison of Bateman gradients among major taxonomic groups. **Table C.** Predictors of interspecific variation in female Bateman gradients using restricted maximum likelihood (REML) approach. **Table D.** Estimates of polyandry and mating system classification (low-polyandry versus high-polyandry). **Table E.** Global tests of sexual selection in females excluding sex-role reversed species. **Table F.** Determinants of sexual selection in females excluding sex-role reversed species. **Table G.** Global tests of sexual selection in females excluding humans. **Table H.** Determinants of sexual selection in females excluding humans.
(PDF)

**S2 Text. List of primary studies.**
(PDF)

## Acknowledgments

We are very grateful to the authors of all primary studies for making their research accessible, especially those providing additional data allowing us to compute Bateman gradients. We also thank Jonathan Henshaw and David Shuker for discussions and comments on previous versions of the manuscript. Julie Collet and Karoline Fritzsche kindly provided unpublished data to quantify the level of polyandry of their study organisms.

## Author Contributions

**Conceptualization:** Lucas Marie-Orleach, Tim Janicke.

**Data curation:** Salomé Fromonteil, Lucas Marie-Orleach, Lennart Winkler, Tim Janicke.

**Formal analysis:** Salomé Fromonteil, Tim Janicke.

**Funding acquisition:** Tim Janicke.

**Investigation:** Salomé Fromonteil, Lucas Marie-Orleach, Lennart Winkler, Tim Janicke.

**Methodology:** Salomé Fromonteil, Tim Janicke.

**Project administration:** Tim Janicke.

**Visualization:** Salomé Fromonteil, Tim Janicke.

**Writing – original draft:** Salomé Fromonteil, Tim Janicke.

**Writing – review & editing:** Lucas Marie-Orleach, Lennart Winkler.

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
