## [Editor Report · Decision Letter 0]

3 Aug 2022

Dear Dr Janicke, 

Thank you for submitting your manuscript entitled "Sexual selection in females and the evolution of polyandry" for consideration as a Research Article by PLOS Biology.

Your revised manuscript has now been evaluated by the PLOS Biology editorial staff, and I'm writing to let you know that we would like to send your submission out for re-review.

Once your full submission is complete, your paper will undergo a series of checks in preparation for peer review. After your manuscript has passed the checks it will be sent out for review. To provide the metadata for your submission, please Login to Editorial Manager (https://www.editorialmanager.com/pbiology) within two working days, i.e. by Aug 05 2022 11:59PM.

Kind regards,

Roli

Roland Roberts, PhD

Senior Editor

PLOS Biology

rroberts@plos.org

---

## [Decision Letter · Decision Letter 1]

12 Sep 2022

Dear Dr Janicke,

Thank you for your patience while we considered your revised manuscript "Sexual selection in females and the evolution of polyandry" for consideration as a Initial Research Submission at PLOS Biology. Your revised study has now been evaluated by the PLOS Biology editors, the Academic Editor and the original reviewer (reviewer #1). You'll see that we have also solicited input from an additional reviewer (reviewer #2) for a more holistic view of the manuscript. 

In light of the reviews, which you will find at the end of this email, we are pleased to offer you the opportunity to address the remaining points from the reviewers in a revision that we anticipate should not take you very long. We will then assess your revised manuscript and your response to the reviewers' comments with our Academic Editor aiming to avoid further rounds of peer-review, although might need to consult with the reviewers, depending on the nature of the revisions.

**IMPORTANT - SUBMITTING YOUR REVISION**

*Resubmission Checklist*

*Published Peer Review*

*PLOS Data Policy*

*Blot and Gel Data Policy*

Sincerely,

Roli Roberts

Roland Roberts, PhD

Senior Editor

PLOS Biology

rroberts@plos.org

REVIEWERS' COMMENTS:

Reviewer #1:

[identifies herself as Hanna Kokko]

IMPORTANT: See attached file for full formatted review!

Reviewer #2:

In general, I find the revised manuscript to be much more balanced, which still providing evidence that sexual selection on females may be more common than most assume (or at least that the scope for sexual selection is wider) and that this is related to the evolution of multiple mating. The authors have done a thorough job of revising the text with these ideas in mind. The authors still favor Bateman's gradient as a measure of sexual selection (even if incomplete), as is clear in the response to reviewer comments. There may be points where I would quibble with the wording. But all in all, the caveats and limitations are much clearer and are more central to the paper, rather than mentioned but downplayed, as in the previous version.

I particularly appreciate the addition the Box that directly addresses the limitations of the Bateman's gradient. I also appreciate the additional analysis after removal of sex-role reversed species, and the color coding by taxonomic group in Figure 2.

A few small editorial comments:

Line 178: Given the possibility for feedbacks (e.g. mate choice copying by females, selection for fecundity by males), the phrasing "is related to" is preferable to "depends on". 

Line 293: "less biased" would be clearer

Line 328: To avoid misreading, clarify that you mean mate limitation experienced by females (competition for males), as opposed to limitation in females

---

## [Editor Report · Decision Letter 2]

2 Nov 2022

Dear Dr Janicke,

Thank you for your patience while we considered your revised manuscript "Sexual selection in females and the evolution of polyandry" for publication as a Research Article at PLOS Biology. This revised version of your manuscript has been evaluated by the PLOS Biology editors and the Academic Editor.

Based on our Academic Editor's assessment of your revision, we are likely to accept this manuscript for publication, provided you satisfactorily address the following data and other policy-related requests.

IMPORTANT: Please attend to the following:

a) Please change your Title to something more declarative and informative (and more appealing for a wider readership). We suggest "Sexual selection is widespread in females and plays a key role in the evolution of animal mating systems" (taken directly from your Abstract).

b) Thank you for the detailed data provision in Github. However, because this can be changed at any time, we need an immutable DOI'd version of record to be provided, for example in Zenodo.

c) Please cite the location of the data clearly in all relevant main and supplementary Figure legends, e.g. “The data underlying this Figure can be found in https://doi.org/XXXX”

We expect to receive your revised manuscript within two weeks. 

*Published Peer Review History*

*Press*

Sincerely,

Roli Roberts

Roland Roberts, PhD

Senior Editor,

rroberts@plos.org,

PLOS Biology

DATA NOT SHOWN?

---

## [Editor Report · Decision Letter 3]

14 Nov 2022

Dear Tim,

Thank you for the submission of your revised Research Article "Sexual selection in females and the evolution of polyandry" for publication in PLOS Biology. On behalf of my colleagues and the Academic Editor, Gail Patricelli, I'm pleased to say that we can in principle accept your manuscript for publication, provided you address any remaining formatting and reporting issues. These will be detailed in an email you should receive within 2-3 business days from our colleagues in the journal operations team; no action is required from you until then. Please note that we will not be able to formally accept your manuscript and schedule it for publication until you have completed any requested changes.

Sincerely,

Roli

Senior Editor

PLOS Biology

rroberts@plos.org